# *Salmonella* spp., *Escherichia coli* and *Enterobacteriaceae* Control at a Pig Abattoir: Are We Missing Lairage Time Effect, Pig Skin, and Internal Carcass Surface Contamination?

**DOI:** 10.3390/foods12152910

**Published:** 2023-07-31

**Authors:** Rui Dias Costa, Vanessa Silva, Ana Leite, Margarida Saraiva, Teresa Teixeira Lopes, Patrícia Themudo, Joana Campos, Madalena Vieira-Pinto

**Affiliations:** 1CITAB—Centre for the Research and Technology of Agro-Environmental and Biological Sciences/Inov4Agro—Institute for Innovation, Capacity Building and Sustainability of Agri-Food Production, University of Trás-os-Montes and Alto Douro, Quinta de Prados, 5000-801 Vila Real, Portugal; joanacampos@utad.pt; 2CECAV—Animal and Veterinary Research Centre, University of Trás-os-Montes and Alto Douro, Quinta de Prados, 5000-801 Vila Real, Portugal; van-silva@live.com.pt (V.S.); aleite@utad.pt (A.L.); 3INSA—National Institute of Health Dr. Ricardo Jorge, Food Microbiology Laboratory, Reference Unit, Department of Food and Nutrition, Rua Alexandre Herculano 321, 4000-055 Porto, Portugal; margarida.saraiva@insa.min-saude.pt (M.S.); teresa.lopes@insa.min-saude.pt (T.T.L.); 4INIAV—National Institute of Agrarian and Veterinary Research, Bacteriology and Micology Laboratory, Avenida da República, Quinta do Marquês, 2780-157 Oeiras, Portugal; patricia.themudo@iniav.pt; 5Veterinary Science Department—Gab. B.1.02, University of Trás-os-Montes and Alto Douro, Quinta de Prados, 5000-801 Vila Real, Portugal

**Keywords:** *Enterobacteriaceae*, *Escherichia coli*, *Salmonella*, abattoir, pig skin, faecal contamination, internal carcass surface, external carcass surface

## Abstract

To provide meat safety and consumer protection, appropriate hygiene control measures at an abattoir are required. This study aimed to evaluate the influence of visual fecal contamination level (VFCL) and lairage time (LT) on pig skin (PS) and external (ECS) and internal (ICS) carcass surfaces. The presence of *Enterobacteriaceae*, *Escherichia coli* (*E. coli*) and *Salmonella* in PS, ECS, and ICS were evaluated. A total of 300 paired samples were collected from 100 pigs. Results underlined the importance of the skin (*Enterobacteriaceae*: 3.27 ± 0.68 log CFU/cm^2^; *E. coli*: 3.15 ± 0.63 log CFU/cm^2^; *Salmonella*: 21% of samples) as a direct or indirect source of carcass contamination. Although VFCL revealed no significant effect (*p* > 0.05), the increase of LT had a significant impact (*p* < 0.001) on *Enterobacteriaceae* and *E. coli* levels across all analysed surfaces, and *Salmonella* presence on ICS (*p* < 0.01), demanding attention to LT. Also, the ICS showed a higher level of these bacteria compared to ECS. These results highlight the need of food business operators to consider ICS as an alternative area to sample for *Salmonella*, as a criterion for process hygiene based on EC Regulation No. 2073/2005, and as a potential contamination source to be integrated in the hazard analysis critical control point (HACCP) plans.

## 1. Introduction

According to the United Nations Food and Agriculture Organization (FAO) pork is the most consumed meat in the world (34%) [1], making up a substantial part of the diet of most people, offering excellent nutrition due to its high protein content and with plenty of vitamins and minerals [2]. Each year, over 23 million individuals become ill due to the consumption of contaminated food, as reported by the World Health Organization (WHO) in 2017. Animal-based food products such as meat, namely pork, poultry, beef, and sheep were found to be the source of approximately 60% of all foodborne illnesses in the same year [3]. Consumption of food with pathogenic bacteria causes a large number of diseases with significant effects on human health and the economy [4]. The majority of pathogenic microorganisms that may be present in pork originate from the gastrointestinal tract and skin, and their occurrence can be linked to direct or indirect contamination during the slaughtering process [5]. In fact, Hoek et al. [6] reported that pigs with *Salmonella* on their skin cause contamination at the slaughter line. The influence of skin microflora on the microbial quality of final pig carcasses in abattoir operations is more complex compared to cattle slaughter. This complexity arises from the successive changes in skin microflora during various processing stages such as scalding, dehairing, singeing, polishing, and washing. Nevertheless, studies have demonstrated the presence of foodborne pathogens, including *Salmonella*, on both the skin of pigs before slaughter and the resulting carcasses. This highlights a direct correlation between skin contamination of live pigs prior to stunning and the subsequent contamination of carcasses [7]. 

For pig carcasses, the European Commission (EC) Regulation No. 2073/2005, and its subsequent amendments, stipulates that the count of *Enterobacteriaceae* and detection of *Salmonella* spp. must be used as a process hygiene criterion (PHC) [8]. If the values on contaminated carcasses are higher than the limit set by this EC Regulation, corrective actions must be taken by the food business operator (FBO) not only to improve hygiene practices during slaughtering, but also to review process controls. According to Barco et al. [9], *Enterobacteriaceae* and *E. coli* are two interchangeable hygienic indicators and *Escherichia coli* (*E. coli*) is utilized to specifically assess the degree of fecal contamination (FC). In Australia the analysis of the level of *E. coli* on carcass surfaces during refrigeration is used as a critical control point in pig abattoirs [10]. In addition, the Australian Meat Standard imposes that the meat companies have to confirm their carcass-refrigeration processes [10]. However, in Europe currently, *E. coli* is not used as a PHC in pig carcasses. Since microbiological methods used for *E. coli* counts are quicker and less expensive than *Enterobacteriaceae*, studies should be developed to analyze its potential as a fecal indicator in pig skin and carcass surface. 

In the European Union (EU), salmonellosis is the second most reported zoonotic disease in humans. In 2021, pork accounted for a significant proportion (31.1%) of reported cases [11,12]. The reported cases were most prevalent in Belgium, Cyprus, Finland, France, Ireland, Italy, Poland, and Sweden, whereas disease attribution to hen laying and pigs are similar in the Netherlands [12]. 

Tonsils, ileum, ileocolic, and mandibular lymph nodes, as well as pig faeces can be important sources of carcass contamination in the slaughter stages by *Salmonella* spp. [13,14,15]. If the lairage time (LT) is extended, the presence of this bacteria may increase, as was referred to by Morgan et al. [16].

According to the EC Regulation No. 2073/2005, sampling for *Salmonella* analysis on ECS typically involves the use of an abrasive sponge sampling method. However, this regulation specifies that it should be prioritized in the areas with the highest probability of contamination for sampling. Knowing that the intestinal gut is considered the primary source of fecal carcass contamination, a hypothesis was formulated to investigate whether the ICS exhibited higher contamination levels compared to the ECS. If this hypothesis is true, and since the ICS is not currently analyzed, it could potentially hinder the implementation of effective corrective measures aimed at reducing human salmonellosis cases associated with pork consumption.

Hence, the main goals of this study included the evaluation of the visual fecal contamination level (VFCL) prior to slaughter and LT in the occurrence of *Enterobacteriaceae*, *E. coli*, and *Salmonella* on carcass, and the comparison of the level of contamination of the external and internal surface of pig carcasses.

## 2. Materials and Methods

The present study was conducted at an abattoir with a horizontal layout located in the northern region of Portugal. The facility conducts pig slaughter three times a week and is equipped to handle a daily throughput of up to 250 pigs. The technological process of the slaughter involves several stages, including carbon dioxide stunning, bleeding, vertical scalding, shaving, first wash/scraping, singeing, second wash/scraping, evisceration, final wash, and refrigeration, as depicted in Figure 1. 

### 2.1. Data Collection

During the sampling period for each pig, the LT before slaughter in hours and VFCL were recorded. The LT was determined by analysing the information recorded by the FBO. Due to logistic restrictions, VFCL was evaluated after the stunning phase and before bleeding. To assess VFCL, the external half carcass surface was divided into four main areas. These areas were further subdivided into four sub-areas each scored with a value of 0.25 if VFCL was observed. The final VFCL level could range from 0 to 4 values (Figure 2). An informed consent statement was obtained from the food business operator (the pig owner) involved in the study. This study complied with the Declaration of Helsinki maintaining anonymity and existing informed consent of all participants.

### 2.2. Sample Size and Sampling Procedures

To determine the counts and/or presence of *Enterobacteriaceae*, *E. coli*, including *E. coli* O157, and *Salmonella*, a total of 300 paired samples (swabs) were collected from 100 carcasses: PS after stunning; ICS and ECS before refrigeration. To collect these samples, sterile sponge-swabs hydrated with tryptone salt broth (Biokar Diagnostics, Allonne, France) were used. The abrasive pad method was applied according to the guidelines of ISO 17604 [17], covering a total of 1000 cm^2^ from the hindquarter downward to the forequarter.

Each swab was placed in a separate, sterilized container, properly identified, and transported under refrigeration conditions to the laboratory within two hours. The sampling procedures followed the guidelines outlined in EC Regulation No. 2073/2005 [8].

### 2.3. Microbiological Analyses

In the laboratory, each sample was aseptically transferred into stomacher bags after adding 20 mL of salt tryptone (Biokar Diagnostics, Allonne, France) to each bag. The samples were homogenized in a stomacher for 90 seconds (Figure 3).

#### 2.3.1. *Enterobacteriaceae* and *E. coli* Counts 

The quantification of *Enterobacteriaceae* and *E. coli* in each of the pig samples was conducted following the procedures specified in ISO 21528-2:2004 [18] and ISO 16649-2:2001 [19], respectively. The results were reported as log colony forming units (CFU)/cm^2^ (Figure 3).

#### 2.3.2. Isolation and Identification of *E. coli* O157

The determination of the presence of *E. coli* O157 was conducted following the guidelines set out in ISO 16654:2001 [20]. The sorbitol-negative colonies from MacConkey Sorbitol Agar (CT-SMAC) (Liofilchem, Roseto degli Abruzzi, Italy), which were indole-positive, urea-negative, and oxidase-negative, were subjected to a latex agglutination test using the *E. coli* O157 Test Kit (Oxoid, Basingstoke, UK). The colonies that showed a negative result in the agglutination test were sent to the National Institute of Health Dr. Ricardo Jorge (INSA) reference laboratory (Porto, Portugal) for genetic identification.

In the INSA reference laboratory, the genetic analysis of *E. coli* O157 was performed. The presence of genes *stx*_1_ and *stx*_2_ was determined using the primers and time-temperature conditions described in multiplex PCR (Table 1). The *rfbO157* gene was detected by simplex PCR (Table 1) [21], using AmpliTaq DNA polymerase (Applied Biosystems, Waltham, USA) and the thermocycler, T100 Thermal Cycler (Bio-Rad, Hercules, USA). Positive and negative controls, as well as an internal amplification control, were used for all PCR reactions. The PCR fragments were separated using 2% agarose gel electrophoresis for 55 min. The amplified fragments were compared to known molecular weight markers after viewing on the GelDoc 2000 transilluminator (Bio-Rad, Hercules, CA, USA) (Figure 3).

#### 2.3.3. Isolation and Identification of *Salmonella*


The detection of *Salmonella* was carried out following the ISO 6579:2002 guidelines [22]. The serotyping of *Salmonella* isolates was conducted using the Kauffmann–White scheme at the National Institute of Agrarian and Veterinary Research (INIAV, Lisbon, Portugal) that is the Portuguese Reference Laboratory for *Salmonella* (Figure 3).

### 2.4. Statistical Analysis

All analyses were conducted in triplicate and the results are presented as mean ± standard deviation (SD). The counts of *Enterobacteriaceae* and *E. coli* in both skin and corresponding carcass samples were determined as CFU)/cm^2^, converted into log CFU/cm^2^, and used to calculate the mean values. The difference significance between groups was assessed using the Student *t*-test. A Pearson correlation analysis was conducted to examine the relationship between the levels of *Enterobacteriaceae* and *E. coli* on pig skin and carcasses, as well as between these indicators and LT and VLFC. The differences in the proportions of positive *Salmonella* samples were determined using the chi-square test. Odds ratios (OR), and their corresponding 95% confidence intervals (CI) were calculated. Statistical analyses were performed using IBM SPSS Statistics software (Version 27, Chicago, IL, USA). Variables with a *p*-value ≤ 0.05 were considered significant.

## 3. Results and Discussion

### 3.1. Enterobacteriaceae, E. coli Counts and Salmonella on Pig Skin

Overall, the average level of *Enterobacteriaceae* and *E. coli* contamination observed on pig’ skin was 3.27 ± 0.68 log CFU/cm^2^ and 3.15 ± 0.63 log CFU/cm^2^, respectively. *Salmonella* was present in 21% (21/100) of the analyzed pig skin. *E. coli* O157 was absent in all samples (Table 2).

Similar but higher results for *Enterobacteriaceae* were previously observed by Blagojevic et al. [7] and Walia et al. [23]. However, caution should be taken when directly comparing the results, as the swab areas employed in the studies conducted by these authors (1500 cm^2^ and 100 cm^2^, respectively) differ from the swab area utilized in our study (1000 cm^2^). According to our knowledge, studies of *E. coli* count on pig skin have not been performed.

Moreover, our findings showed a lower prevalence of *Salmonella* on PS compared to the study conducted by Blagojevic et al. [7] in abattoir A (28%) and abattoir B (40%), which considered these values relatively high in both cases. 

These results underline the importance of the skin as a source of contamination for the abattoir and, consequently, for the direct or indirect contamination of carcasses. In fact, Gill [24] previously discussed this topic and highlighted the impact of enteric pathogens found in animal faeces at the time of slaughter on meat contamination.

#### 3.1.1. Influence of Pig Skin on Carcass Contamination 

The mean levels of *Enterobacteriaceae* and *E. coli* on PS were significantly higher (*p* < 0.001) than that obtained on both surfaces of the carcass (Table 2). A significant positive correlation (*p* < 0.001) was observed between these bacteria on all analyzed surfaces, indicating that a higher presence of bacteria on the skin corresponds to a greater presence on the carcass. This finding aligns with the research conducted by Rossel et al. [25], which suggested a direct link between carcass contamination and the presence of *Salmonella* on the skin of live pigs prior to stunning.

These results, highlights the importance of the technological process of slaughter in reducing its contamination, already remarked by Zdolec et al. [26].

In this study, there was a significant correlation (*p* < 0.001) between the levels of *E. coli* and *Enterobacteriaceae* counts on the three surfaces tested. Similar results were previously found by Salmela et al. [27] on sheep carcasses. However, currently in Europe, *E. coli* is not used as a PHC in pig carcasses. Since microbiological methods used for *E. coli* counts are quicker and less expensive than *Enterobacteriaceae*, the results from this study point out for the alternative use of *E. coli* as a fecal indicator on pig skin and carcass surface.

As on PS, *E. coli* O157 was also not detected on ECS and ICS, which is in accordance with studies previously carried out by Bouvet et al. [28,29], Lenahan et al. [10], and Choi et al. [30]. As with our study, other studies [31,32,33] obtained a high number of false positive results that grew in CT-SMAC. We concluded, as they did, that this medium does not prove to be adequate to efficiently select *E. coli* O157.

The overall percentage of *Salmonella* positive samples was 11% (33/300). Of the 300 samples analyzed, *Salmonella* was detected in 21 (21%) PS, in 10 (10%) ICS and in 3 (3%) ECS (Table 2). The same reduction pattern was reported by Hoek et al. [6]. Also, the three serovars identified in our study were the monophasic variant of *Salmonella* Typhimurium, *S.*
1,4,[5],12:i:- (39.4%), *Salmonella* Rissen (39.4%), and *Salmonella* Derby (15.2%) (Table 3). According to the last report of the European Food Safety Agency (EFSA) [11], *S.* 1,4,[5],12:i:-, is one of the most relevant *Salmonella* serotypes causing human salmonellosis (Top 3) in Europe, including the main reported serotype recovered from pigs and pork meat. In fact, in this study, *S.* 1,4,[5],12:i:- was the most prevalent and identified on both pig skin (8/21; 38.1%) and the internal surface (5/9; 55.6%), confirming that pigs are the main animal reservoir for the monophasic variant of *S.* Typhimurium. Although *S.* Derby and *S.* Rissen have been predominant serotypes in pig and pork meat in Europe, they have a lesser impact on human salmonellosis cases [34]. Interestingly, *S.* Rissen is considered a clinically-relevant serotype, particularly in southern European countries, including Portugal, the same strains in humans, pigs, and products thereof being frequently detected [34]. The authors would like to underline that although *S.* Typhimurium has been most frequently found on pigs in the abattoir [6,11], it was not detected in the present study. 

A higher number of PS-*Salmonella* positives (21%, 21/100) was observed compared to ECS (3%, 3/100) (Table 2). However, by analyzing the data shown in Table 3, it is possible to verify that the three ECS *Salmonella*-positive samples were not associated with *Salmonella*-positive pig skin, equating other hypotheses of contamination in addition to the direct one (skin). 

In the opposite way, from the nine ICS *Salmonella*-positive samples, four (44.4%) had the same serotype identified in the corresponding skin samples. In three of these cases, *S*. 1,4,[5],12:1:- was the serotype identified. How is it possible for Salmonella to be present on the skin and also contaminate the internal surface of the carcass? Two potential explanations are proposed here. First, the contamination may occur after the splitting step when the internal surfaces are exposed to skin contaminants, primarily through indirect contact with equipment, instruments (such as knives), hands, and other sources. Second, this explanation is related to fecal contamination of the internal surfaces during evisceration. This occurs when *Salmonella* from the intestinal contents, which may be the same strain as those present on the skin, contaminates the internal surfaces. It is known that pigs can become orally infected with *Salmonella* from fecal material in the environment prior to slaughter, and simultaneously become soiled with this material, thereby contaminating their skin [25,35]. Furthermore, prior to entering an abattoir, pigs may already carry *Salmonella* on their skin. Despite implementing rigorous hygiene measures during carcass processing, the possibility of cross-contamination to both *Salmonella* positive and negative carcasses cannot be ruled out [6]. Therefore, it is crucial to prioritize efforts to better understand the several potential sources that may contribute to the contamination of both carcass surfaces. This is essential to enable the implementation of robust control measures that can effectively address the risk of contamination.

#### 3.1.2. Influence of Visible Fecal Contamination Level 

The average VFCL observed on pig skin after stunning was 0.4. Considering that the final VFCL level could range from 0 to 4 values, it may be assumed that, in general, the analysed carcasses were not very dirty. This could be related to the fact that before stunning, the animals were submitted to a cleaning shower. These results are in line with EC Regulation No. 853/2004 that defines that the animals must be clean before slaughtered [36].

Statistical analysis showed that there was no significative correlation between VFCL (*p* > 0.05) and the level of *Enterobacteriaceae*, *E. coli*, and presence of *Salmonella* on PS. Hence, it appears that the shower employed for cleansing the pigs, prior to stunning, successfully reduced visible fecal contamination. However, it seems insufficient for the elimination of the specific bacteria under investigation, as they persist on the pig’s skin (Table 2). According to Belluco et al. [37], the process of washing does not necessarily result in a substantial reduction in microbial contamination. This is because bacteria can still adhere to the skin even after washing. Additionally, there is a possibility that the washing procedure itself could potentially lead to a redistribution of bacteria across the surface of the skin. Taking these results into consideration, the FBO should understand that despite slaughtering “clean” (washed) animals, bacteria can remain on the skin, so it should not lighten the implementation of good hygiene practices in order to minimize their spread and cross-contamination.

Based on our findings in this abattoir, it is not recommended to rely solely on the VFCL observed after stunning as an indicator of the presence of the specific indicator bacteria. In contrast to other animals, such as lamb Hauge et al. [38], sheep Byrne et al. [39], and chickens Barco et al. [40], no studies were found associating the visible fecal soiling with carcass contamination in pigs. Based on the authors’ knowledge, this is the first study looking at the relationship between microbiological and VFCL on pig carcasses, proving to be an interesting topic to explore. This knowledge gap has been previously acknowledged in a comprehensive literature review conducted by Barco et al. [9]. The use of this hygiene indicator before showering could be more useful in future studies.

### 3.2. External Carcass Surface vs. Internal Carcass Surface

The average level of *Enterobacteriaceae* and *E. coli* contamination on ICS was 1.65 ± 0.90 log CFU/cm^2^ and 1.34 ± 0.89 log CFU/cm^2^, respectively. Similar values were found by Vieira-Pinto et al. [41]. 

The average level of *Enterobacteriaceae* and *E. coli* contamination on ECS was 0.29 ± 0.52 log CFU/cm^2^ and 0.33 ± 0.58 log CFU/cm^2^, respectively. For *Enterobacteriaceae* lower contamination load was found by Morgan et al. [42], Vieira-Pinto et al. [41], Zweifel et al. [43], Matsubara [44], Pearce and Bolton [45], Spescha et al. [46]; Lindblad [47], Ghafir and Daube [48], and Lenahan et al. [10]. Regarding *E. coli*, similar results, but not directly comparable, were found by Ghafir and Daube [48], Wong et al. [49] and Lindblad [47]. 

The results from this study also showed that the mean level of *Enterobacteriaceae* and *E. coli* on the ICS was significantly higher (*p* < 0.001) than that obtained on ECS. 

*Salmonella* was detected in nine (9%) ICS and in three (3%) ECS, despite not having observed a significant difference (*p* > 0.05). Diverse results regarding the presence of *Salmonella* on ECS were found in the literature from 0%, Wong et al. [49], to 34.9%, Zhou et al. [50]. Furthermore, comparing both surfaces the identification was possible of an additional nine carcasses contaminated with *Salmonella* on ICS (Table 3). The results of the study performed by Hoek et al. [6] are in line with the present study that ICS presents more *Salmonella* than ECS. 

Overall, these findings highlight the need for increased attention toward the ICS, which is not currently subject to mandatory control measures specified in EC Regulation 2073/2005 [51] and subsequent amendments. According to this Regulation, the presence of *Salmonella* spp. on pig carcasses is considered a process hygiene criterion for controlling contamination during the slaughter process. However, this criterion has been revised by EC Regulation No. 217/2014 [52], based on EFSA opinion [11], which recognizes *Salmonella* as a significant risk to public health in relation to the consumption of pig meat. The regulation recommends strengthening the process hygiene criterion for *Salmonella* on pig carcasses. Under the revised criteria, satisfactory control is achieved if the presence of *Salmonella* is detected in a maximum of 3 out of 50 samples from 10 consecutive sampling session. Comparing our results to this EU process hygiene criterion, we can conclude that the process hygiene for external surfaces (3/100) was considered satisfactory, whereas, for internal surfaces (9/100), it did not meet the desired criteria. This underscores the importance of addressing the contamination of ICS to ensure food safety and to comply with the revised process hygiene standards. Although sampling for *Salmonella* analyses as the process hygiene criterion usually used in the ECS areas most likely to be contaminated can be also selected for this purpose (EC Regulation No. 2073/2005) [8]. This could be the case of ICS. For that reason, bearing in mind the results of this study, FBO, may consider ICS as an alternative area to be sampled for *Salmonella* as a process hygiene criterion operator, in order to control the process hygiene and adequately contribute to the reduction of human salmonellosis cases attributed to pork consumption.

Furthermore, if the contamination level of the ICS is higher than the ECS, the FBO must be alert to potential sources of contamination in order to implement effective measures to mitigate this problem, which may include the reduction of cross contamination [41,52,53], as well as the good hygiene practices during evisceration [54].

### 3.3. Effect of the Lairage Time 

The average lairage time (LT) observed in this study was 27 h (27 ± 15). According to EC Regulation No. 853/2004, the animals must be slaughtered without unnecessary delay [36]. This high value might have been impacted by the fact that sample collections were conducted on Mondays. The pig may become more contaminated with bacteria if the LT is extended. The achieved results showed that increasing LT leads to an increase of the level of *Enterobacteriaceae* and *E. coli* both on pig skin and in the respective carcass surfaces, these relationships being significant (*p* < 0.001). In the case of the presence of *Salmonella* on ICS, a significant relationship was also observed with the increase of LT (*p* > 0.01). In the study performed by Morgan et al. [16], it was observed that with the increasing of the LT, the rate of *Salmonella* isolation from caeca and carcass surfaces increased considerably. Hurd et al. [55] also concluded that the LT is a significant risk factor for *Salmonella* contamination and infection in pork. In the work performed by Duggan et al. [56] LT also influenced cross-contamination with *Salmonella*.

These results suggest that FBO must implement hygiene improvements in lairage to assure better efficiency in reducing the levels of *Enterobacteriaceae*, *E. coli* and presence of *Salmonella*. 

## 4. Conclusions

With the present study we identified important results that should be considered by a food business operator when implementing microbiological hazard control measures in a pig abattoir. Of these, we highlight the importance of the skin which, despite being macroscopically “clean”, represents an important potential source of *Salmonella* and other *Enterobacteriaceae* bacteria to the abattoir and to carcass contamination (direct or indirect). Moreover, the lairage time was found to have a significant impact on higher levels of contamination, raising awareness to the food business operator to reduce this period, which is in line with the EC Regulation No. 853/2004 [36]. Furthermore, this study alerts the food business operator to the internal surface of the carcass, relevant under the implementation of EC Regulation No. 2073/2005 and the HACCP-based procedures or other meat hygiene control measures in place. 

## Figures and Tables

**Figure 1 foods-12-02910-f001:**
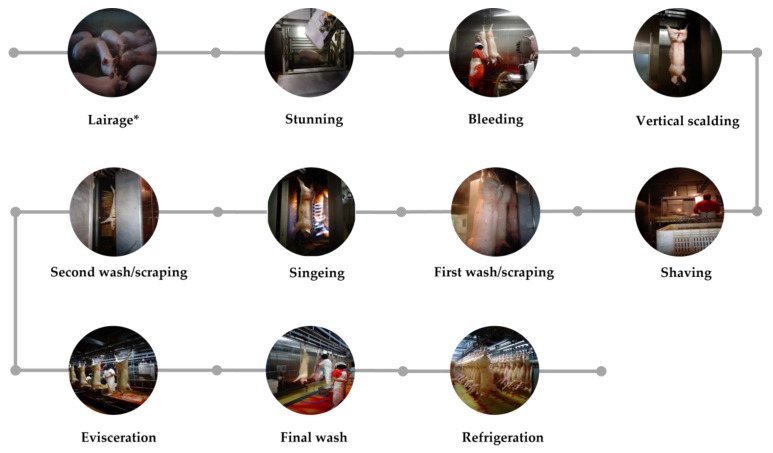
Technological process of the investigated abattoir. * The animals were subjected to showering before being moved to stunning.

**Figure 2 foods-12-02910-f002:**
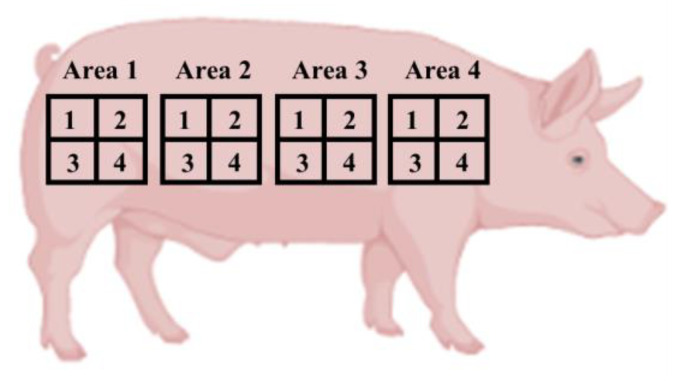
Evaluation scheme of the VLFC.

**Figure 3 foods-12-02910-f003:**
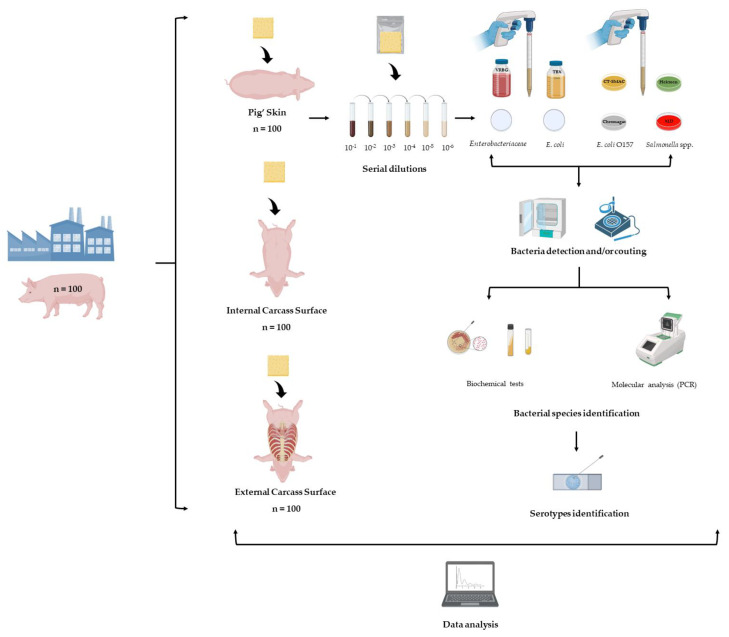
Schematic representation of the methodology used in this study. Enumeration of *Enterobacteriaceae* (ISO 21528-2:2004): VRBG, Violet Red Bile Glucose agar; enumeration of *Escherichia coli* (ISO 16649-2:2001): TBX, Tryptone Bile X-Glucuronide; detection of *E. coli* O157 (ISO 16654:2001): CT-SMAC, Cefixime Tellurite Sorbitol MacConkey; CHROMagar Chromogenic Culture Media; detection of *Salmonella* (ISO 6579): Hektoen, Hektoen Enteric agar; XLD, Xylose Lysine Deoxycholate agar; PCR, Polymerase Chain Reaction.

**Table 1 foods-12-02910-t001:** Sequence of primers used in multiplex PCR and simplex PCR.

Primers ^a^	Sequence (5′ 3′)	Amplicon Size ^b^	Reference
*stx*_1_ (F)	ATA AAT CGC CAT TCG TTG ACT AC	180 bp	[21]
*stx*_1_ (R)	AGA ACG CCC ACT GAG ATC ATC		
*stx*_2_ (F)	GGC ACT GTC TGA AAC TGC TCC	255 bp	
*stx*_2_ (R)	TCG CCA GTT ATC TGA CAT TCT G		
O157(F)	CGG ACA TCC ATG TGA TAT GG	259 bp	
O157(R)	TTG CCT ATG TAC AGC TAA TCC		

^a^ F, forward; R, reverse. ^b^ bp, base pair.

**Table 2 foods-12-02910-t002:** *Enterobacteriaceae* and *E. coli* counts, and presence of *E. coli* O157 and *Salmonella* spp. on different pig surfaces.

Family/Species	PS (Log CFU/cm^2^)	ICS (Log CFU/cm^2^)	ECS (Log CFU/cm^2^)
*Enterobacteriaceae*	3.27 ± 0.68 ^a^	1.65 ± 0.90 ^b^	0.29 ± 0.52 ^c^
*E. coli*	3.15 ± 0.63 ^a^	1.34 ± 0.89 ^b^	0.33 ± 0.58 ^c^
*E. coli* O157	ND	ND	ND
*Salmonella* spp.	21% (21/100)	9% (9/100)	3% (3/100)

The values are represented as mean ± SD (*n* = 3). Different letters indicate significantly different results. PS, Pig Skin; ICS, Internal Carcass Surface; ECS, External Carcass Surface; ND, Not Detected.

**Table 3 foods-12-02910-t003:** *Salmonella* serovars identified on pig skin and on carcass surfaces.

Pig	PS	ICS	ECS
1	*Salmonella* Rissen	-	-
3	*Salmonella* Rissen	-	-
4	*Salmonella* Rissen	-	-
5	*Salmonella* Rissen	-	-
9	Inconclusive ^a^	-	-
27	-	-	*Salmonella* Rissen
28	-	*Salmonella* Rissen	-
34	-	-	*Salmonella* Rissen
39	Inconclusive ^a^	-	-
45	*Salmonella* 1,4,[5],12:i:-	*Salmonella* Rissen	-
46	*Salmonella* 1,4,[5],12:i:-	-	-
47	-	*Salmonella* Rissen	-
48	*Salmonella* Rissen	-	-
51	*Salmonella* Rissen	-	-
52	*Salmonella* Rissen	-	-
54	*Salmonella* Rissen	-	-
56	*Salmonella* Derby	-	-
57	*Salmonella* Derby	-	-
58	*Salmonella* Derby	*Salmonella* Derby	-
60	-	-	*Salmonella* Derby
86	*Salmonella* 1,4,[5],12:i:-	-	-
94	*Salmonella* 1,4,[5],12:i:-	*Salmonella*1,4,[5],12:i:-	-
95	-	*Salmonella* 1,4,[5],12:i:-	-
96	*Salmonella* 1,4,[5],12:i:-	-	-
97	*Salmonella* 1,4,[5],12:i:-	-	-
98	-	*Salmonella* 1,4,[5],12:i:-	-
99	*Salmonella* 1,4,[5],12:i:-	*Salmonella* 1,4,[5],12:i:-	-
100	*Salmonella* 1,4,[5],12:i:-	*Salmonella* 1,4,[5],12:i:-	-

PS, Pig Skin; ECS, External Carcass Surface; ICS, Internal Carcass Surface; -, serotype not identified. ^a^ Inconclusive serotype identification due to a rough form of *Salmonella* strain.

## Data Availability

The data used to support the findings of this study can be made available by the corresponding author upon request.

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
