# Peer review of "Salmonella spp., Escherichia coli and Enterobacteriaceae Control at a Pig Abattoir: Are We Missing Lairage Time Effect, Pig Skin, and Internal Carcass Surface Contamination?"

_foods, 2023, doi:10.3390/foods12152910_

Round 1

Reviewer 1 Report

A nicely done study that provides ideas on better ways to monitor for pathogens in pig meat processing. Various specific comments and corrections are suggested below.

I suggest a modification of the title to make it clearer. I thought to change “surface internal carcass contamination” to “ carcass internal surface…” since readers may get confused of what is being meant. If you prefer the original version that is fine since the meaning is clear in the paper.

Salmonella spp., Escherichia coli and Enterobacteriaceae control at pig abattoirs: Are we missing the lairage time effect and pig skin and carcass internal surface contamination?

Pig’s skin/carcasses etc. (or pig’ ….) can be generically written as pig skin and so forth.

Line 33 … attention to LT.

Lairage has seen as a safety risk for a long time. Wouldn’t the attention be not on lairage time but rather lairage site management and animal hygiene before slaughter not just the duration?

Line 35-46 …and as an effective contamination critical control point. Also lines 357-360.

The way this sentence is written it is assumed this sampling site can represent a useful CCP. I think this is perhaps a little premature without sufficient surveying and where applicable meta-analysis is done. This is due to the need to understand the effects of ICS versus ECS contamination for cross-contamination and the associated quantifiable risk. I suggest rewriting the sentence as

“…and as a potential contamination source that could be considered for integration into HACCP plans”.

Lines 80-85 I suggest rewriting this section for improved clarity and to also define the hypothesis better.

Lines 348-349 … external surfaces (3/100) the process hygiene was considered satisfactory in opposite to internal surfaces (9/100).

Is it truly opposite? If you to repeat the experiment in another abattoir would you expect the same finding? I think this result should come with qualifications until more evidence is available.

Minor corrections

Line 24 To provide meat safety and consumer protection, …

Lines 26-27 …. pig skin (PS) and carcass contamination including that of external (ECS) and internal (ICS) carcass surfaces.

Line 30 …Salmonella was present in 21% of samples)…

Line 32 … lairage time influenced significantly increased (p < 0.001)…

Line 50 ….the gastrointestinal tract and skin…

Lines 54-55 The statement made here should be rewritten to improve clarity. Opposite to what (cattle?) and comparing what (skin versus viscera as sources of contamination)?.

Line 56 For pig carcasses, the European…

Line 70-72 Shorten and combine the sentences here:

 …, salmonellosis ranks as the second most reported zoonosis in humans with pork contributing a high proportion (31.1%) of cases in the EU in 2021 [10]. Cases were most prevalent in Belgium,..

Line 90 … external and internal surface of pig carcasses.

Line 157-158 Mention the model of the thermocycler used.

Line 201. What were the values from Blagojevic et al., [6]? Also given counts are expressed per square centimetre why wouldn’t the data be comparable? A smaller swab surface area could be less sensitive but if enough samples were taken this wouldn’t matter.

Line 249 …for the monophasic variant of S. Typhimurium.

Line 269 A higher number of PS-Salmonella positives….

Lines 270-271 However, by analyzing the data shown in Table 3, it is possible to verify that the three ECS Salmonella-positive samples…

Line 275 …identified in the corresponding skin samples.

Lines 276-277 The question framed here needs to be rewritten.

How can Salmonella be present on the skin but also contaminate the internal surface of the carcass? Two possible explanations are provided here.

Line 278 … when internal facies become…

e.g. facies interna

Line 288 …it is  important to try to better understand the complexity…

Lines 335-336 … allowed identification of an additional 7 carcasses contaminated with Salmonella.

Lines 339-340 …the analysis for which is not mandated under EC Regulation 339 2073/2005 (and subsequent amendments).

Overall the English writing is good. There were some sentences that need to be looked at again and rephrased to make the expression and message clear. I have identified some if not all of these sentences.

Author Response

Dear Editor of Foods,

In reply to the review performed on the paper entitled " Salmonella spp., Escherichia coli and Enterobacteriaceae control at pig’s abattoir: Are we missing Lairage Time Effect, Pig’ Skin and Internal Carcass Surface Contamination?", we would like to acknowledge the valuable comments performed by the editor that kindly accepted to revise our manuscript.

We would like to confirm that we have addressed most issues and answered the questions that have been made.

We hope the answers below and modifications introduced in the manuscript are clear and concise enough as required by the reviewers, in order to enable the publication of the manuscript in Foods.

Answer to referee’s comments and queries

Detailed responses to Reviewer 1

1º Reviewer´s comment: Line 27; add: ‘internal carcass surface (ICS). The presence of EnterobacteriaceaeEscherichia coli (E. coli)

Our reply: We appreciate the reviewer’s comments and insightful considerations. Thank you for your suggestion, it was already corrected.

2º Reviewer´s comment: Line 41-44; add references for FAO

Our reply: Thank you for your suggestion. As required the reference was added.

3º Reviewer´s comment: Line 62; put the whole name after abbr. Escherichia coli (E. coli)

Our reply: Thank you for your suggestion, it was already changed.

4º Reviewer´s comment: Line 164; Table 1 “Sized of amplicon fragments” in my opinion should be ‘Amplicon size’ and pb should be changed on bp base pair – 180 bp, 255 bp

Our reply: Thank you for your suggestion, it was already changed.

5º Reviewer´s comment: Line 235; Bouvet et al., [27, 28]

Our reply: Thank you for your suggestion, it was already changed.

6º Reviewer´s comment: Line 321; ‘No other references were found about this topic.’ After this sentence, I am sure that this valuable article needs professional English correction.

Our reply: Thank you. The English was revised throughout the manuscript.

Reviewer 2 Report

The article titled Salmonella spp., Escherichia coli and Enterobacteriaceae control at Pig’s Abattoir: Are we missing Lairage Time Effect, Pig’s skin and internal carcass surfers contamination? Is a valuable and well-written paper but needs scientific English correction. I have only minor comments:

Line 27; add: ‘internal carcass surface (ICS). The presence of EnterobacteriaceaeEscherichia coli (E. coli) …….

Line 41-44; add references for FAO

Line 62; put the whole name after abbr. Escherichia coli (E. coli)

Line 164; Table 1 “Sized of amplicon fragments” in my opinion should be ‘Amplicon size’ and pb should be changed on bp base pair – 180 bp, 255 bp

Line 235; ‘ Bouvet et al., [27, 28]

Line 321; ‘No other references were found about this topic.’ After this sentence, I am sure that this valuable article needs professional English correction.

Congrats to the Authors  GOOD JOB

Author Response

Dear Editor of Foods,

In reply to the review performed on the paper entitled " Salmonella spp., Escherichia coli and Enterobacteriaceae control at pig’s abattoir: Are we missing Lairage Time Effect, Pig’ Skin and Internal Carcass Surface Contamination?", we would like to acknowledge the valuable comments performed by the editor that kindly accepted to revise our manuscript.

We would like to confirm that we have addressed most issues and answered the questions that have been made.

We hope the answers below and modifications introduced in the manuscript are clear and concise enough as required by the reviewers, in order to enable the publication of the manuscript in Foods.

Answer to referee’s comments and queries

Detailed responses to Reviewer 2

1º Reviewer´s comment: A nicely done study that provides ideas on better ways to monitor for pathogens in pig meat processing. Various specific comments and corrections are suggested below. I suggest a modification of the title to make it clearer. I thought to change “surface internal carcass contamination” to “ carcass internal surface…” since readers may get confused of what is being meant. If you prefer the original version that is fine since the meaning is clear in the paper.

Salmonella spp., Escherichia coli and Enterobacteriaceae control at pig abattoirs: Are we missing the lairage time effect and pig skin and carcass internal surface contamination?

Pig’s skin/carcasses etc. (or pig’ ….) can be generically written as pig skin and so forth.

Our reply: Thank you for your suggestion. We revise all the manuscript, and we didn’t find a sentence with “surface internal carcass contamination” to change as suggested. Hence, we didn’t comply this request.

2º Reviewer´s comment: Line 33 … attention to LT. Lairage has seen as a safety risk for a long time. Wouldn’t the attention be not on lairage time but rather lairage site management and animal hygiene before slaughter not just the duration?

Our reply: Thank you for your comment, with which we agree. Indeed, management and conditions are very important as a risk factor but also the length of stay in the lairage. If pigs are asymptomatic carriers of Salmonella, a prolonged period of time favored faecal excretion of this agent and, consequently, contamination of the skin. In addition, faecal-oral infection of other animals may be promoted, especially at the level of the tonsils and mesenteric lymph nodes, with consequent risk of cross contamination during slaughter process. In this, if reviewer agrees, we would like to maintain the sentence as it is.

3º Reviewer´s comment: Line 35-46 …and as an effective contamination critical control point. Also lines 357-360. The way this sentence is written it is assumed this sampling site can represent a useful CCP. I think this is perhaps a little premature without sufficient surveying and where applicable meta-analysis is done. This is due to the need to understand the effects of ICS versus ECS contamination for cross-contamination and the associated quantifiable risk. I suggest rewriting the sentence as“…and as a potential contamination source that could be considered for integration into HACCP plans”.

Our reply: Thank you for your suggestion, it was already changed.

4º Reviewer´s comment: Lines 80-85 I suggest rewriting this section for improved clarity and to also define the hypothesis better.

Our reply: Thank you. The section was rewritten as suggested.

5º Reviewer´s comment: Lines 348-349 … external surfaces (3/100) the process hygiene was considered satisfactory in opposite to internal surfaces (9/100).

Is it truly opposite? If you to repeat the experiment in another abattoir would you expect the same finding? I think this result should come with qualifications until more evidence is available.

Our reply: Thank you. This sentence was written based on what the results of this study suggest and not on other abattoirs. Each abattoir may have different practices and therefore different results.

6º Reviewer´s comment: Line 24 To provide meat safety and consumer protection, …

Our reply: Thank you for your suggestion, it was already changed.

7º Reviewer´s comment: Lines 26-27 …. pig skin (PS) and carcass contamination including that of external (ECS) and internal (ICS) carcass surfaces.

Our reply: Thank you for your suggestion, it was already changed.

8º Reviewer´s comment: Line 30 …Salmonella was present in 21% of samples)…

Our reply: Thank you for your suggestion, it was already changed.

9º Reviewer´s comment: Line 32 … lairage time influenced significantly increased (p < 0.001)…

Our reply: Thank you for your suggestion. In order to better understand the sentence, it was changed as suggested. 

10º Reviewer´s comment: Line 50 ….the gastrointestinal tract and skin…

Our reply: Thank you for your suggestion, it was already changed.

11º Reviewer´s comment: Lines 54-55 The statement made here should be rewritten to improve clarity. Opposite to what (cattle?) and comparing what (skin versus viscera as sources of contamination)?.

Our reply: Thank you for your suggestion, it was already changed. The modifications were made in order to better understand the transmitted idea.

12º Reviewer´s comment: Line 56 For pig carcasses, the European…

Our reply: Thank you for your suggestion, it was already changed.

13º Reviewer´s comment: Line 70-72 Shorten and combine the sentences here:

  …, salmonellosis ranks as the second most reported zoonosis in humans with pork contributing a high proportion (31.1%) of cases in the EU in 2021 [10]. Cases were most prevalent in Belgium,..

Our reply: Thank you for your suggestion, it was already changed.

14º Reviewer´s comment: Line 90 … external and internal surface of pig carcasses.

Our reply: Thank you for your suggestion, it was already changed.

15º Reviewer´s comment: Line 157-158 Mention the model of the thermocycler used.

Our reply: Thank you for your suggestion, it was already changed.

16º Reviewer´s comment: Line 201. What were the values from Blagojevic et al., [6]? Also given counts are expressed per square centimetre why wouldn’t the data be comparable? A smaller swab surface area could be less sensitive but if enough samples were taken this wouldn’t matter.

Our reply: Thank you for the comment. We believe that results could be different when swab surface are also different (For example 10 times less for Walia et al.). So, we need to be prudent during comparison. For this reason, if the reviewer agrees we would like to re-write the sentence as: “However, caution should be taken if direct comparison is made because the swab area used by these authors (1500 cm2 and 100 cm2, respectively) was different from ours (1000 cm2)."

17º Reviewer´s comment: Line 249 …for the monophasic variant of S. Typhimurium.

Our reply: Thank you for your suggestion, it was already changed.

18º Reviewer´s comment: Line 269 A higher number of PS-Salmonella positives….

Our reply: Thank you for your suggestion, it was already changed.

19º Reviewer´s comment: Lines 270-271 However, by analyzing the data shown in Table 3, it is possible to verify that the three ECS Salmonella-positive samples…

Our reply: Thank you for your suggestion, it was already changed.

20º Reviewer´s comment: Line 275 …identified in the corresponding skin samples.

Our reply: Thank you for your suggestion, it was already changed.

21º Reviewer´s comment: Lines 276-277 The question framed here needs to be rewritten. How can Salmonella be present on the skin but also contaminate the internal surface of the carcass? Two possible explanations are provided here.

Our reply: Thank you for your suggestion, it was already changed.

22º Reviewer´s comment: Line 278 … when internal facies become… e.g. facies

Our reply: Thank you for your suggestion, but we prefer to maintain the words according to FAO reports. 

23º Reviewer´s comment: Line 288 …it is  important to try to better understand the complexity…

Our reply: Thank you for your suggestion. The alterations were made to improve the sentence.

24º Reviewer´s comment: Lines 335-336 … allowed identification of an additional 7 carcasses contaminated with Salmonella.

Our reply: Thank you. The sentence was improved.

25º Reviewer´s comment: Lines 339-340 …the analysis for which is not mandated under EC Regulation 339 2073/2005 (and subsequent amendments).

Our reply: Thank you for your suggestion, it was already changed.